# Immune Cell Networks Uncover Candidate Biomarkers of Melanoma Immunotherapy Response

**DOI:** 10.3390/jpm12060958

**Published:** 2022-06-11

**Authors:** Duong H. T. Vo, Gerard McGleave, Ian M. Overton

**Affiliations:** 1The Patrick G Johnston Centre for Cancer Research, Queen’s University Belfast, 97 Lisburn Road, Belfast BT9 7AE, UK; hvo01@qub.ac.uk (D.H.T.V.); gmcgleave01@qub.ac.uk (G.M.); 2Health Data Research Wales and Northern Ireland, Queen’s University Belfast, 97 Lisburn Road, Belfast BT9 7AE, UK

**Keywords:** immune checkpoint, melanoma, ovarian carcinoma, systems immunology, network biology, immunotherapy, precision oncology, biomarker, nivolumab, systems medicine

## Abstract

The therapeutic activation of antitumour immunity by immune checkpoint inhibitors (ICIs) is a significant advance in cancer medicine, not least due to the prospect of long-term remission. However, many patients are unresponsive to ICI therapy and may experience serious side effects; companion biomarkers are urgently needed to help inform ICI prescribing decisions. We present the IMMUNETS networks of gene coregulation in five key immune cell types and their application to interrogate control of nivolumab response in advanced melanoma cohorts. The results evidence a role for each of the IMMUNETS cell types in ICI response and in driving tumour clearance with independent cohorts from TCGA. As expected, ‘immune hot’ status, including T cell proliferation, correlates with response to first-line ICI therapy. Genes regulated in NK, dendritic, and B cells are the most prominent discriminators of nivolumab response in patients that had previously progressed on another ICI. Multivariate analysis controlling for tumour stage and age highlights CIITA and IKZF3 as candidate prognostic biomarkers. IMMUNETS provide a resource for network biology, enabling context-specific analysis of immune components in orthogonal datasets. Overall, our results illuminate the relationship between the tumour microenvironment and clinical trajectories, with potential implications for precision medicine.

## 1. Introduction

The immune system functions to eliminate tumour cells; however, immunoediting can result in the immune response promoting cancer progression [1]. The ‘equilibrium’ and subsequent ‘escape’ phases of immunoediting involve a selective microenvironment where cancer cells with the capacity to evade the immune response become dominant. Therefore, tumour-associated leukocytes can adopt a range of different cellular programmes that may either impede or contribute to cancer progression [2]. Therapeutic activation of antitumour immunity has had a huge clinical impact, particularly in producing long-term remission [3]. Immune checkpoint inhibitors (ICIs) exploit the CTLA-4 and PD-L/PD-L1 pathways to activate T lymphocytes in the tumour microenvironment [3]. ICIs are effective in a range of cancers, including melanoma, renal cell carcinoma, and non-small cell lung cancer [4]. However, current forms of ICI therapy are also associated with multiple side effects, including autoimmune reactions [5]. A response rate varies between cancer types, and a minority of patients typically respond to single-agent treatment [6]. Factors that correlate with ICI response include high mutational burden, a T cell-inflamed microenvironment and the formation of tertiary lymphoid structures with CD20^+^ B-cells [7,8]. A deeper understanding of the factors that drive the heterogeneity of clinical response to ICI therapy could inform patient-specific regimes in order to enhance outcomes and reduce side effects [9]. Indeed, treatment efficacy may be enhanced by combination therapy; for example, by targeting multiple pathways or by inhibiting tumour plasticity alongside ICIs [10,11]. An early example of precision oncology measures HER2/neu to inform Trastuzumab treatment in breast cancer [12]; similarly, the development of companion biomarkers is a key step in maximising the patient benefit from ICI therapy.

Immunotherapy has been particularly successful in melanoma, a common form of skin cancer associated with a high mortality rate due to its propensity to metastasise [13]. However, many melanoma patients do not benefit from sustained responses to ICIs, with median progression of 12 months or less [14,15,16]. Characterisation of the molecular networks that control resistance and response to ICI therapy in poor prognosis cancers may provide mechanistic insight towards new therapeutic tools and ultimately inform prescribing decisions. Network models provide a useful abstraction of complex biological systems [17] and may enhance the development of methods for risk stratification in precision oncology [18]. We mapped the activation and differentiation of five different immune cell types in order to identify molecular correlates of clinical immunotherapy responses by analysing data from the Immune Response In Silico (IRIS) study [19] and melanoma patients treated with a PD-1 inhibitor (nivolumab) [20,21,22]. Our results delineate immunological processes at the genome-scale and propose candidate immunotherapy response biomarkers that have prognostic value in independent cohorts of melanoma patients [23].

## 2. Results

We produced IMMUNETS, a set of five immune cell coregulation networks derived from the Immune Response In Silico (IRIS) transcriptome data, representing cells from healthy human donors [19]. The IRIS study captured immune transcriptomes across multiple activated and differentiated cell states and so provides a basis for modelling the networks of genes involved in immune regulation and immunotherapy response. Our approach (Figure 1) utilised transcriptome data from IRIS, melanoma patients treated with immunotherapy [20,22] and The Cancer Genome Atlas (TCGA) [23]. IMMUNETS provided for the derivation of immune cell-specific focus networks (Figure 2, Appendix A) and investigation of immune regulation in the context of nivolumab response (Figure 3). The resulting candidate biomarkers showed clinically relevant expression differences in an independent dataset (Figure 4).

### 2.1. IMMUNETS: Modelling Immune Cell Differentiation and Activation

We built IMMUNETS with data from the IRIS study, which assayed key immune cell types in multiple states of activation and differentiation [19]. There are five different IMMUNETS networks, each focused upon a broad class of immune cells (T cells, B cells, NK cells, Monocytes, and Dendritic cells). Each network connects genes that are both differentially regulated in immune cell function and are highly correlated within the cell grouping for which the network was constructed. Differentially expressed genes were identified either between or within immune cell types and were used to construct correlation networks for T cells (T-net: 3069 genes, 54,108 edges), Natural Killer (NK) cells (NK-net: 5944 genes, 65,426 edges), B cells (B-net: 4626 genes, 105,349 edges), Monocytes (Mono-net: 2867 genes, 55,150 edges) and Dendritic cells (Dend-net: 5071 genes, 58,995 edges). Together, the five correlation networks cover 9668 genes in total, of which 3262 are found in only one network and 2820 are present in exactly two of the five networks (Table 1, Appendix A). The representation scope of IMMUNETS is defined by regulation in the 214 samples assayed by IRIS for the five immune cell networks [19], and our network inference protocol. For example, the secreted cytotoxic proteases termed granzymes are key effectors of cytotoxicity found in T-net (GZMB, GZMM, GZMH) and NK-net (GZMA, GZMB, GZMK, GZMM). GZMA and GZMK are present in NK-net, which aligns with single-cell analysis of NK cells [24]; however, these genes are also important in T cell functions [25]. Genes found only in one of the five IMMUNETS networks have highly coordinated expression changes in IRIS for the corresponding cell grouping, but the genes may also be expressed in other cell types. Therefore, a gene is only represented in IMMUNETS if it shares a pattern of expression with at least one additional gene across the cell states assayed in IRIS. IRIS contains a further 14 samples for plasma cells; however, the protocol applied here did not produce a network for these samples, and therefore plasma cells are not represented in IMMUNETS.

NK-net is the largest overall (2748 genes) and has the greatest overlap with the other networks, for example, sharing 751 and 535 genes with Dend-net and B-net, respectively. B-net has the most unique genes (*n* = 1022) and the highest average degree (22.8). We derived focus networks for each immune cell grouping by taking the IMMUNETS genes given in Table 1 as input for the NetNC algorithm and using HumanNet as the reference network [26,27] (Figure 2, Appendix A). As expected, the focus networks have connections between clusters that are annotated with biologically related terms. For example, the T cell focus network (Figure 2A) connects clusters for ‘cell cycle process’, ‘DNA repair’, ‘DNA replication’, and ‘RNA metabolic process’; similarly, there are connections in the B cell network between ‘ion transport’ and ‘transmembrane transport’ clusters.

The T cell focus network covers biological processes required for growth and proliferation (cell cycle, RNA metabolism, translation), including key T cell proliferation genes (Figure 2A). Indeed, proliferation is central to normal T cell biology [28]. A ‘regulation of lymphocyte proliferation’ (RegLP) cluster has seven genes that were found only in the T-net correlation network. The RegLP cluster includes molecules associated with p38 MAP kinase signal transduction (MAPK11, MAP2K6) [29] and cell surface glycoproteins CD8 and CD28 that have well-known roles in antigen-induced activation [30]. Other cell surface glycoproteins and interleukins identified in the RegLP cluster are important for a range of T cell functions, for example, the following: CD7 is a differentiation marker for CD8+ T cells [31]; CD5 downregulation potentiates T-cell antitumour activity [32]; IL4, IL13, IL5 locate in a cytokine gene cluster on chromosome five and regulate multiple immune cell functions, including in T helper 2 cells [33,34]; IL17A is produced by activated T cells where it may impact many processes, notably promoting tumour progression [35,36]. Signalling through CD3 is important for T cell activation [37]. However, we note that CD3 subunits are found in multiple immune cell types in IMMUNETS, including T cells, but do not appear in the T cell focus network, which only contains genes from T-net and at most one other IMMUNETS network. In particular, CD3E and CD3EAP are represented in three IMMUNETS networks and therefore were not input to NetNC for the T cell focus network generation. Most of the genes in the T cell focus network (76%, 141/185) are coordinately regulated within other immune cell types in IMMUNETS, especially NK cells. On the other hand, approximately half of the genes in the B cell focus network (Figure 2B) are found exclusively in the B-net correlation network (51%, 409/797). These B cell-regulated genes cover a wide range of cell activities, including ABC transporters, ADAM proteases, cytochrome P450 family, growth factor receptors, solute transport, and many others (Appendix A). A ‘B cell activation’ cluster contains genes involved in B cell antigen receptor complex (BCR) signaling; for example, CD79A, CD79B, CD19, CD22, CD20 (MS4A1), and transcription factors important for B cell development, including BCL11A, PAX5 [38,39,40,41,42,43]. An ‘inflammatory response’ cluster contains IL-20 subfamily interleukins IL22 and IL22RA1, IL20RA, and IL10RB, which form receptor complexes that bind IL22 or IL26 [44]. IL22 is important for B cell recruitment to tertiary lymphoid structures [45], and depletion of B cells reduces IL22 production, which can stimulate cell behaviour typical of aggressive tumours [46,47]. Therefore, the production of IL22 by B cells might be part of a positive feedback loop driving poor prognosis in immunoedited tumours.

### 2.2. IMMUNETS Genes Stratify Melanoma Patients by Response to Nivolumab

Genes found in at most two of the five IMMUNETS cell types were investigated as candidate biomarkers in a cohort of advanced melanoma patients [20] (Figure 3). Data were analysed from tumour biopsies taken before treatment with nivolumab, an inhibitor of PD-1 [20,21]. The following two cohorts were considered: patients previously treated with ipilimumab that had progressed (MEL_PROG, *n* = 26) and those that were ipilimumab-naive (MEL_NAI, *n* = 23). Both cohorts were part of the NCT01621490 trial, which took patients that were refractory, intolerant to, or had refused standard therapy [48]. MEL_NAI and MEL_PROG, respectively, had a total of 30 and 136 differentially expressed IMMUNETS genes between nivolumab responders and non-responders (Mann-Whitney test *q* < 0.05, 2-fold change). Therefore, we identified differentially regulated immune cell genes that correlate with response to nivolumab in vivo and stratified treatment response in unsupervised clustering (Figure 3). A ‘good response’ cluster emerged in MEL_NAI and in MEL_PROG, respectively, containing 0/6 and 2/8 patients with progressive disease (PD). Similarly, a ‘poor response’ cluster in each cohort, respectively, has 1/17 and 0/18 patients with either a complete or partial response. We reasoned that immunotherapy response mechanisms would likely be represented within IMMUNETS genes that are upregulated in patients who responded to nivolumab. In contrast, a range of biological processes which drive or correlate with tumour aggressiveness could be upregulated in non-responders, and so appear a less attractive pool for the discovery of candidate immunotherapy response biomarkers. Therefore, we focus on genes upregulated in tumours that respond well to therapy because we expect that these are more pertinent to molecular control of therapy response.

Immune tolerance is mediated by the expression of PD-L1 in many cell types and PD-1 in T cells, which is therefore critical in preventing autoimmune disease [49]. While nivolumab binds to and inactivates PD-1 [21], PD-L1 expression is a more informative predictor of clinical response than PD-1 [50,51]. PD-1 and its ligand PD-L1 did not pass the criteria for inclusion in IMMUNETS and therefore do not appear in Figure 3. The genes found only in the T-net network and upregulated in the MEL_NAI nivolumab response group are CENPI and PIP5K1A. Upregulation of the CENPI gene that functions in cell proliferation typically confers poor prognosis in colorectal cancer [52,53]. Therefore, elevated CENPI levels in nivolumab-responsive patients likely reflect expression in tumour-associated immune cells; indeed, proliferation is integral to an effective T cell immune response [54]. This correlation of immune proliferation with immunotherapy response is suggestive of pre-existing immune activation that is enhanced by immunotherapy, aligning with reports that responsive patients have inflamed or ‘immune hot’ tumours [55]. PIP5K1A functions in phosphatidylinositol signalling, which impacts multiple cell processes, including T cell activation [56]. Thus, the elevation of PIP5K1A in nivolumab responders is also consistent with ICI treatment acting to enhance pre-existing T cell inflammation. Our results also highlight genes regulated in B cell and dendritic cell biology that correlate with response to nivolumab in treatment-naive melanoma. For example, the cytokine GREM1 was found only in B-net and upregulated in the MEL_NAI treatment response group. GREM1 is a BMP antagonist and VEGF agonist and is downregulated in human B cells exposed to the toxin TCDD, which causes immunosuppression in addition to other harmful effects [57,58]. The presence of G protein gamma subunit GNG4 in the B-net and upregulation with response to therapy suggests a role in B cell activation, consistent with the importance of G-protein coupled receptor signalling in the humoural immune response [59]. NRP2 is one of several genes found in Dend-net that are upregulated in the nivolumab response group. While NRP2 is expressed in multiple immune cell types, it is important for dendritic cell maturation, migration, and T cell activation [60]. Overall, our results in MEL_NAI align with reports that ‘immune hot’ status correlates with response to ICIs [55] and suggest candidate nivolumab response biomarkers.

A favourable treatment response in MEL_PROG correlates with upregulation of CD247, the zeta chain of CD3, which forms a complex with the T cell receptor (TCR) and is central to the T cell immune response [37,61]. CD247 is expressed in a wide range of leukocytes [62] and is regulated in dendritic cells as well as NK cells in IMMUNETS. CD8α, produced from CD8A, is a canonical marker for the T cell population involved in tumour surveillance and forms a heterodimer with CD8β, or may homodimerise [63]; CD8A is the only gene exclusive to T-net that correlates with nivolumab response in MEL_PROG. Genes that correlate with MEL_PROG therapy response and are found in only one IMMUNETS cell type include the following: ADAM28 from B-net, which controls lymphocyte transendothelial migration [64]; the hydroxysteroid 11β dehydrogenase type 1 (HSD11B1) from Mono-net that regulates the resolution of the inflammatory response, including in macrophages, and correlates with CD4+ T cell activation [65]; IL33 from Dend-net that stimulates CD8+ T cell antitumour responses and downregulates PD-1 [66]. Therefore, our analysis with IMMUNETS suggests that dendritic cells could be a source of IL33 in antitumour immunity, although IL33 is also produced by other cell types [67]. Several genes upregulated in the MEL_PROG good response group can drive tumour aggressiveness when expressed in cancer cells. For example, ADAM28 expression in tumour epithelium is associated with poor prognosis, but is also important for T cell mobilisation to metastatic lesions [68,69]. Therefore, upregulation of otherwise poor prognosis genes in the good response group suggests that their correlation with ICI therapy response arises from regulation within immune cells rather than within tumour epithelial cells.

DAVID analysis of the immune-regulated genes from IMMUNETS that are differentially expressed between therapy response groups in MEL_NAI and MEL_PROG (Table 2, Appendix A) was consistent with results from NetNC (Figure 2) and the heatmap (Figure 3). Broadly, MEL_NAI has significant functional clusters for processes involved in cell proliferation, while the differences in treatment response for MEL_PROG are associated with immune regulation, including T cell signalling and activation. As noted above, the proliferation genes that correlate with a good response to treatment in MEL_NAI are regulated in IMMUNETS and therefore may reflect immune cell proliferation in the tumour. The difference between the mechanisms of therapy response in the two cohorts is underlined by the four IMMUNETS genes significantly changed in both cohorts. These are the following: HOMER1, a scaffolding and signal transduction protein that regulates T cell activation [70,71]; CENPI, a centromere protein that is essential for mitosis [72]; DEPDC1, a cancer-related gene that is required for cell cycle progression [73]; TRIP13, a regulatory protein involved in mitosis and DNA repair [74]. Interestingly, these shared genes have opposite expression patterns between response and non-response groups in MEL_NAI and MEL_PROG; for example, HOMER1 correlates with good response in MEL_NAI and poor response in MEL_PROG (Figure 3).

### 2.3. Investigation of Candidate Nivolumab Response Biomarkers Expression in an Independent Cohort

We investigated candidate nivolumab response biomarkers identified in IMMUNETS and regulated in MEL_NAI and MEL_PROG in independent melanoma cohorts (Figure 4). Gene expression data were available from melanoma biopsies for patients receiving nivolumab, some of whom had previously received ipilimumab [22]; defining two cohorts, VALID_NAI (immunotherapy-naïve, *n* = 30) and VALID_PROG (immunotherapy progressed, *n* = 21). The differentially expressed IMMUNETS genes from MEL_NAI and MEL_PROG were respectively evaluated in VALID_NAI and VALID_PROG. HOMER1 is the only gene differentially expressed in both MEL_NAI and VALID_NAI, although with an opposite relationship to treatment response. In MEL_NAI, HOMER1 expression was higher in nivolumab responders, but higher in non-responders for VALID_NAI. HOMER1 has multiple splice variants with distinct roles, for example in neurological studies where overexpression of HOMER1A or HOMER1C respectively blunt or enhances response to cocaine [75,76,77]. This splice variation might explain the difference in the relationship of HOMER1 expression values to nivolumab response in MEL_NAI and VALID_NAI, due to overlapping probes for HOMER1A/HOMER1C. All of the candidate biomarkers from MEL_PROG that were validated in VALID_PROG had the same expression pattern with respect to nivolumab response. Nivolumab response genes that were significant in MEL_PROG and validated in VALID_PROG function in B cell and T cell biology as well as immune cell recruitment. For example, the ADAM28 metalloproteinase binds to α4β1 integrin and enhances lymphocyte adhesion to endothelial layers and is important for T cell mobilisation [64,68,69]. Validated B cell genes that correlate with therapy response include the following: TENT5C, a non-canonical poly(A) polymerase that is highly expressed in activated B lymphocytes [78]; CD79A, a component of the BCR [79]; CIITA, a transcriptional activator that induces MHC gene expression [80,81]; IKZF3, a transcription factor that regulates B cell differentiation and function [82]. Validated T cell genes that correlate with therapy response were as follows: IL2RB, an IL-2 receptor subunit [83]; CD247, which forms part of the T cell receptor–CD3 complex [84] and CD4, a T-cell surface glycoprotein that functions as a coreceptor for antigen-MHC class II complex on T cells [85]. Two kinases, CABYR and MAG2, were downregulated in nivolumab responders in both MEL_PROG and VALID_PROG, as follows: CABYR is a phosphorylation-dependent calcium-binding protein expressed in naïve and memory B cells [86]; MAGI-2 is a scaffolding protein and membrane-associated guanylate kinase [87] that interacts with the tumour suppressor PTEN [88].

### 2.4. Candidate Immune Biomarkers Risk Stratify Melanoma by Overall Survival

We sought to identify candidate ICI response biomarkers from genes that are coregulated in IMMUNETS because they may capture the status of tumour-associated immune cells, which is the primary substrate for ICI therapy. For this purpose, we took the IMMUNETS genes that were differentially expressed in treatment response groups in both the discovery and validation datasets (BIO_13). The prognostic value of BIO_13 was explored in melanoma TCGA data (*n* = 390, MEL_TCGA) [23] with univariate analysis of patient groups defined using unsupervised clustering of BIO_13 expression values (Table 3). Following correction for multiple hypothesis testing [89], 10/13 genes were significant in MEL_TCGA (log-rank *q* < 0.05). Recruitment criteria ensured that MEL_TCGA comprised only patients with no previous systemic treatment, excepting adjuvant interferon ≥ 90 days prior, and previous work reports a correlation between immune activation and good prognosis in this cohort [23]. In agreement with these findings, ten selected IMMUNETS genes that are upregulated in both MEL_PROG and VALID_PROG therapy response groups also portend a good prognosis in our analysis of MEL_TCGA (Table 3, Appendix A). BIO_13 was further validated in a metastatic melanoma cohort (*n* = 174, MIXED_ICI) who received either single or combined immune checkpoint inhibitors and was drawn from four separate studies [90,91,92,93]. MIXED_ICI includes diverse primary tumour sites, for instance, the uveal tract and skin are primary tumour sites in the Samuel et al. study [90]. Patients in MIXED_ICI also received several different immune checkpoint inhibitors; for example, in the Gide et al. study, 63 patients received anti-PD1 immunotherapy (Nivolumab/Pembrolizumab) and 57 patients were treated with combined anti-PD1 and anti-CTLA-4 (Ipilimumab with Nivolumab/Pembrolizumab) [93]. Six of the BIO_13 genes (ADAM28, CD247, IKZF3, CIITA, CD79A, IL2RB) were significant in MIXED_ICI (*q* < 0.05, Table 3). Therefore, we delineate factors that are regulated in immune cells, which correlate with response to ICI therapy and may drive tumour clearance in other melanoma treatment pathways.

Stepwise feature selection with Akaike information criterion (AIC) regularisation was performed for Cox proportional hazards modelling [94,95] of overall survival, taking as input the BIO_13 genes that were significant in univariate analysis, as well as age and tumour stage. A significant top-scoring model (tumour stage, age, CIITA, IKZF3, CD247, TENT5C; likelihood ratio test *p* = 2 × 10^−7^) was returned for the MEL_TCGA training set (*n* = 255, TCGA_TRAIN). In all of the selected features, tumour stage, age, CIITA, and IKZF3 were individually significant in the multivariate model (*p* < 0.05, Appendix A) and taken forward into the four-factor model shown in Table 4, although IKZF3 was not significant. The Cox models satisfy the proportional hazards assumption (Appendix A). The multivariate four-factor model (tumour stage, age, CIITA, IKZF3) was validated in the MEL_TCGA independent validation data (*n* = 135, TCGA_VALID). The high-risk and low-risk groups have strikingly different overall survival in TCGA_VALID (Figure 5, log-rank test *p* = 0.00012). Regularisation with AIC ensures that the selected molecular features add information over and above the clinical variables analysed. As expected, a higher tumour stage and age confer a worse prognosis. CIITA is a transcriptional activator of MHC genes [80,81,96]. MHC molecules play an essential role in activating effector immune cells such as T lymphocytes. CIITA is shared between B-net and Dend-net and correlates with lower risk (HR 0.83) in the multivariate model. It may be a surrogate marker for the activation status of antigen-presenting dendritic cells and B cells in the tumour microenvironment during melanoma progression. In contrast, the expression of IKZF3 encoding the transcription factor Aiolos correlates with a higher risk (HR 1.20) [97]. IKZF3 regulates apoptosis by inducing Bcl-2, inhibits pre-B-cell expansion by suppressing c-Myc expression [98,99] and is constitutively expressed throughout NK-cell ontogeny [100]. While IKZF3 is associated with a higher risk in the TCGA melanoma cohort (Table 4, Appendix A), it also correlates with treatment response in MEL_PROG and VALID_PROG (Figure 3 and Figure 4). These results suggest that IKZF3 marks immune cell populations that both support immunotherapy responses and are characteristic of aggressive tumours.

## 3. Discussion

We present five immune cell networks (IMMUNETS) which model gene coregulation in T cells (T-net), B cells (B-net), NK cells (NK-net), monocytes (Mono-net), and dendritic cells (Dend-net) (Table 1, Appendix A). IMMUNETS provide a resource for understanding the pathways and protein complexes that control immune cell activation and differentiation, including systems immunology approaches [101]. Our analysis is complementary to tissue-based resources such as ImSig [102]. We took IMMUNETS genes that are coregulated in at most two cell types as a basis for the derivation of immune cell focus networks, which capture key biological processes regulated within each immune cell type, including immune-specific gene clusters (Figure 2, Appendix A). We find that regulation of proliferation is a major theme within the T cell focus network and where 76% of genes are overlapping with at least one other cell type in IMMUNETS. In contrast, the majority of genes in the B cell focus network are not covered by another IMMUNETS cell type; a wide range of cell functions found only in the B-net are present in the B cell focus network; BCR complex signalling, for example.

Analysis with IMMUNETS genes revealed correlated nivolumab responses in two advanced melanoma cohorts (Figure 3). The MEL_PROG cohort had progressed on ipilimumab, whereas MEL_NAI had not received prior immune checkpoint therapy [20]. The biological mechanisms underlying nivolumab response differ between the two cohorts, clearly demonstrated in functional clustering (Table 2); MEL_NAI immunotherapy response genes broadly function in proliferation, while the response in MEL_PROG is characterized by immune regulation. Genes that associate with poor prognosis when expressed in epithelial tumour cells, for example, drivers of proliferation, were upregulated in patients who responded well to nivolumab; therefore, these genes most likely mark immune cell proliferation in our analysis. Results evidence differences in the tumour microenvironment for ipilimumab-naive and ipilimumab-progressing patients, as well as in differing ICI response trajectories. Thirteen genes that were both immune-regulated in IMMUNETS and differentially expressed in MEL_NAI or MEL_PROG (BIO_13) were validated in pseudo-matched independent cohorts (VALID_NAI, VALID_PROG) [22]. Twelve genes in BIO_13 were discovered and validated in treatment-progressed melanoma cohorts (MEL_PROG, VALID_PROG respectively); these twelve genes have functions important for leukocyte recruitment, B cell and T cell biology. Reassuringly, validated immune response genes that were upregulated in nivolumab responders include key T cell genes, for example, CD4 and CD247. Additionally, our findings emphasise the importance of B cells in immunotherapy success. For example, a BCR component, CD79A, in BIO_13, was upregulated in nivolumab response. Our results also align with an independent study of a combined anti-CTLA-4 and anti-PD-1 immunotherapy cohort [93] where five of the BIO_13 genes (CD247, ADAM28, CIITA, IKZF3, and CD79A) correlated with immunotherapy response.

Eleven of the BIO_13 genes correlate with the overall survival of TCGA melanoma patients [23] in univariate analysis, in line with the established importance of immune regulation in melanoma [103]. Furthermore, in a mixed immunotherapy cohort (MIXED_ICI), six of the BIO_13 stratified patients with diverse treatments (ADAM28, CD247, IKZF3, CIITA, CD79A, IL2RB). Only 5% of TCGA melanoma patients were identified as stage IV (metastatic), while MIXED_ICI represents metastatic melanoma patients [23,90,91,92,93]. Multivariate regularised Cox regression with stepwise feature selection produced a model containing the variables age, tumour stage, CIITA, and IKZF3 with significant p-values individually (Appendix A), demonstrating added value for these genes over the clinical variables examined, and its significance was validated in a testing set of MEL_TCGA. Higher expression of the CIITA transcription factor correlates with lower risk; CIITA functions to upregulate MHC genes, and so lower CIITA expression might facilitate immune evasion [81,104]. IKZF3 encodes Aiolos, a transcription factor that is important in B lymphocyte development [100]. Increased IKZF3 expression confers higher risk in the multivariate model, which is the opposite to the relationship with survival from univariate analysis (Appendix A). The Programmed Death Ligand 1 (PD-L1) is one of two ligands of the Programmed Cell Death 1 (PD-1) protein, which is expressed in a variety of cells, including tumours, where it functions to modulate immune reactions [105]. PD-L1 is an approved predictive biomarker for immune checkpoint therapy in multiple cancers, including bladder cancer and breast cancer, and also has prognostic significance [106,107,108]. Our analysis excluded PD-L1 in the IMMUNETS network construction step because it did not pass the stringent statistical tests that were designed to select markers specific to the individual immune cell types analysed. PD-L1 was significant in univariate survival analysis with both the MEL_TCGA and MIXED_ICI cohorts (Appendix A). Overall, survival analysis in further independent cohorts aligns with the expected importance of immune genes in melanoma progression [13] and specifically identifies significant prognostic value for six of the BIO_13 genes. Our analysis with IMMUNETS has facilitated the identification of candidate biomarkers in multiple diverse melanoma cohorts and suggests mechanisms whereby different immune cell types in the tumour microenvironment could influence disease progression. Further investigation of candidate biomarkers presented in our study would be valuable across multiple ethnically diverse cohorts in order to advance precision medicine across a breadth of different populations. 

## 4. Methods

### 4.1. Co-Expression Gene Networks and Focus Network construction

The Immune Response In Silico (IRIS) immune cell gene expression data were obtained from the Gene Expression Omnibus database, accession GSE22886 [19]. ANOVA with Benjamini-Hochberg false discovery rate correction was applied to determine significantly differentially expressed genes across the full IRIS dataset (*q* < 0.01) [89,109]. Pearson correlation was calculated separately for each leukocyte dataset and only highly correlated significant gene pairs were retained (r > 0.9, *q* < 0.01). Therefore, the correlation values reflect co-regulation within, rather than between, each of the five cell types. We note that correlation-based distance measures perform well in separating functionally related genes from randomly selected pairs [110]. Genes from each of the five correlation networks for the regulated genes were taken as input to the NetNC algorithm if they were found in no more than two IMMUNETS networks; in order to produce five focus networks. NetNC analysis used the ‘FTI’ setting and HumanNet as the base network [26,27]. The five focus networks output from NetNC-FTI were visualized by Cytoscape and annotated with the BiNGO plugin using a significance threshold of *q* < 0.05 [111,112] (Appendix A), all of the expressed genes in IRIS [19] were taken as the background gene list for enrichment analysis.

### 4.2. Differential Expression Analysis of IMMUNETS Genes in Melanoma Treatment Response

RNA-seq data from melanoma patients who received treatment with nivolumab were obtained as FPKM values from the Gene Expression Omnibus database, accession GSE91061 [20]. Low-expression genes were filtered by using filterByExpr function in R [113]. Filtering low-expression genes has been demonstrated to be essential in analysing RNA-seq data to avoid sampling noise and enhance differentially expressed gene detection sensitivity [114]. Density plots were used to determine the threshold for filtering (Appendix A). Filtering parameters were 1 for minimum count and 25% for minimum proportion. We analysed forty-nine patients with complete clinical data, in the following two cohorts: individuals without prior immune checkpoint inhibitor treatment (MEL_NAI, *n* = 23) and those who had received ipilimumab but had progressed (MEL_PROG, *n* = 26). The response of patients to nivolumab was defined by the Response Evaluation Criteria in Solid Tumors (RECIST) version 1.1. Complete response (CR) and partial response (PR) indicate the elimination or decrease in lesions; stable disease (SD) indicates no significant increase or decrease; progressive disease (PD) represents significant tumour growth [115]. We classified CR and PR as ‘response’ and PD as ‘non-response’; patients with SD were not assigned to either group. The 5898 IMMUNETS genes that were shared by no more than two of the five cell types and measured in the melanoma RNA-seq data [20] were taken forwards into differential expression analysis between the response and non-response groups. Differential expression was assessed separately in MEL_NAI and MEL_PROG (Mann-Whitney test *q* < 0.05, 2-fold change), with appropriate false discovery rate correction [89]. Expression values for the differentially expressed genes were Blom transformed before hierarchical clustering with Euclidean distance and visualisation as a heatmap [116]. Differentially expressed genes for MEL_NAI and MEL_PROG were analysed separately using DAVID with the 5898 IMMUNETS genes (above) as the reference gene list, Functional Annotation Clusters with enrichment score > 1.3 were taken as significant [117].

### 4.3. Validation of Candidate Immunotherapy Response Genes in an Independent Cohort

Significant differentially expressed genes (DEGs) from MEL_NAI and MEL_PROG were validated in an independent advanced melanoma cohort (*n* = 51) [22]. Response to nivolumab was available according to the RECIST [115] criteria. Transcriptome sequencing data was obtained as transcripts per million (TPM) values and low-expression genes had already been excluded in the downloaded data [22]. Patients with no prior exposure to immunotherapy (VALID_NAI, *n* = 30) or who had progressed on ipilimumab (VALID_PROG, *n* = 21) were taken for validation of DEGs in MEL_NAI, MEL_PROG respectively (Mann-Whitney *q* < 0.05, 2-fold change). Validated genes were visualized with Euclidean distance hierarchical clustering on the heatmap after Blom transformation [116].

### 4.4. Evaluation of Candidate Biomarkers for Risk Stratification of Melanoma

For univariate analysis, risk groups were defined by Gaussian mixture modelling (GMM) with unsupervised selection of cardinality [118] using the BIO_13 gene expression values in MEL_TCGA and MIXED_ICI (*n* = 174) [90,91,92,93]. Log-rank test p-values with false discovery rate correction [89] identified significant genes (*q* < 0.05). Multivariate analysis was conducted with genes that were significant in univariate analysis along with tumour stage and age. Tumour stage was coded numerically, translating from the ordinal values of the TNM staging system for melanoma [119]. Stages I, Ia, Ib, and Ic were assigned a value of 1; stages II, IIa, IIb, and IIc a value of 2; stages III, IIIa, IIIb, and IIIc a value of 3; stage IV was assigned a value of 4. MEL_TCGA data was split into training (*n* = 255, TCGA_TRAIN) and validation data (*n* = 135, TCGA_VALID) where the proportion of patients across AJCC tumour stages was held constant between TCGA_TRAIN and TCGA_VALID. We selected features for Cox proportional hazards modelling using stepwise backwards elimination and Akaike Information Criterion (AIC) regularisation [94,95]; the four significant features (Age, stage, CIITA, IKZF3) were taken forwards into a new model trained on TCGA_TRAIN. The proportional hazards assumption was evaluated using the Grambsch-Therneau test [120] (Appendix A). Risk groups in TCGA_VALID were defined by the score threshold that partitioned TCGA_TRAIN into two similarly sized groups (*n* = 128, *n* = 127).

## Figures and Tables

**Figure 1 jpm-12-00958-f001:**
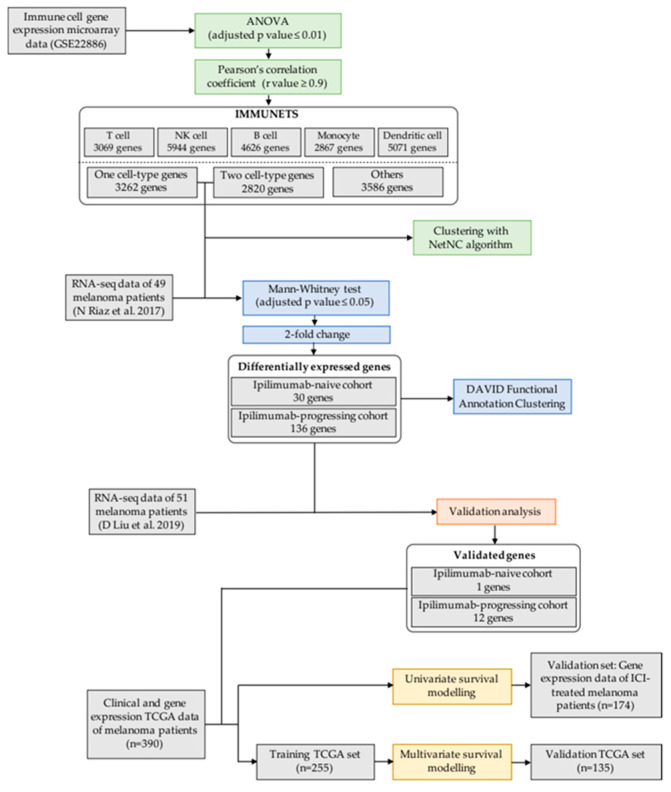
Methodological outline. Networks were constructed from genes that were both highly correlated and differentially expressed across five cell types (IMMUNETS, top). Genes present in at most two of the five IMMUNETS cell networks were input into NetNC, enabling investigation of immune regulation in patients with differing nivolumab response profiles (centre). IMMUNETS genes that correlated with nivolumab response were validated with an independent melanoma dataset (*n* = 51); the significant genes (BIO_13) were further investigated in risk stratification (bottom; TCGA_TRAIN, *n* = 255) taking cohorts independent of prognostic model selection for validation (TCGA_VALID *n* = 135, MIXED_ICI *n* = 174).

**Figure 2 jpm-12-00958-f002:**
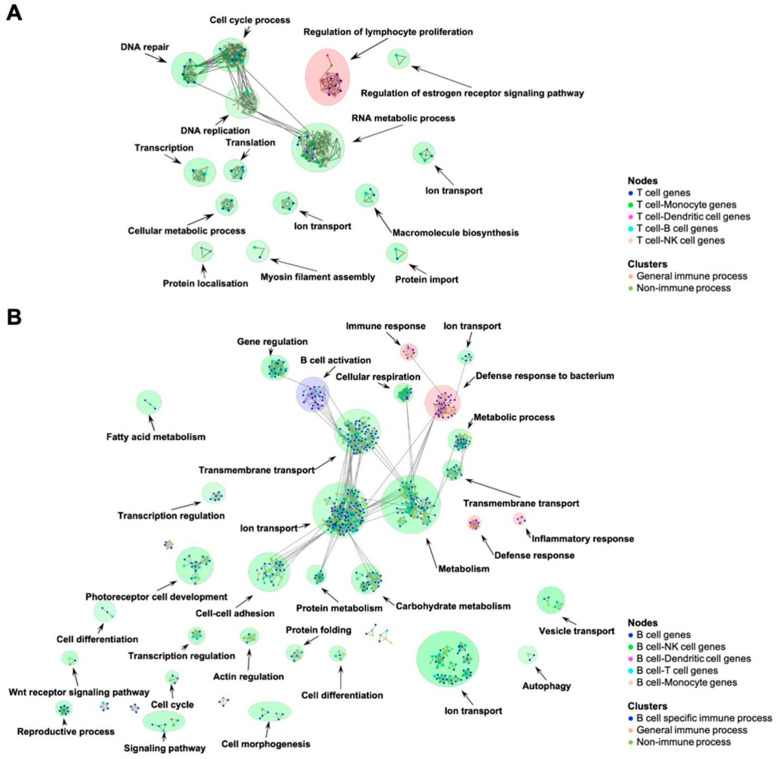
T cell and B cell focus networks. Node colour shows the overlap of genes with other cell types in IMMUNETS and genes present in a single network are shown in blue. Edges were significant according to NetNC-FTI analysis (methods). The coloured circles around clusters indicate broad annotation classes where significant GO terms were identified; red corresponds to general immune GO annotations, blue represents cell-type specific immune processes and green shows clusters with annotation terms that are not immune-specific. (**A**) Most clusters in the T cell focus network represent processes important for cell proliferation. A ‘regulation of lymphocyte proliferation’ cluster (red) contains multiple T cell genes (IL17A, CD5, IL13, IL5, IL2, CD8A, CD28). (**B**) The B cell focus network has one cell-specific immune cluster (blue) ‘B cell activation’, including CD79A, CD79B and CD19. The four clusters annotated with general immune processes (red), such as ‘inflammatory response’, contain important B cell genes for example IL22, IL10RB, IL20RA, and IL22RA1. Cytoscape sessions for the IMMUNETS focus networks are available in Appendix A.

**Figure 3 jpm-12-00958-f003:**
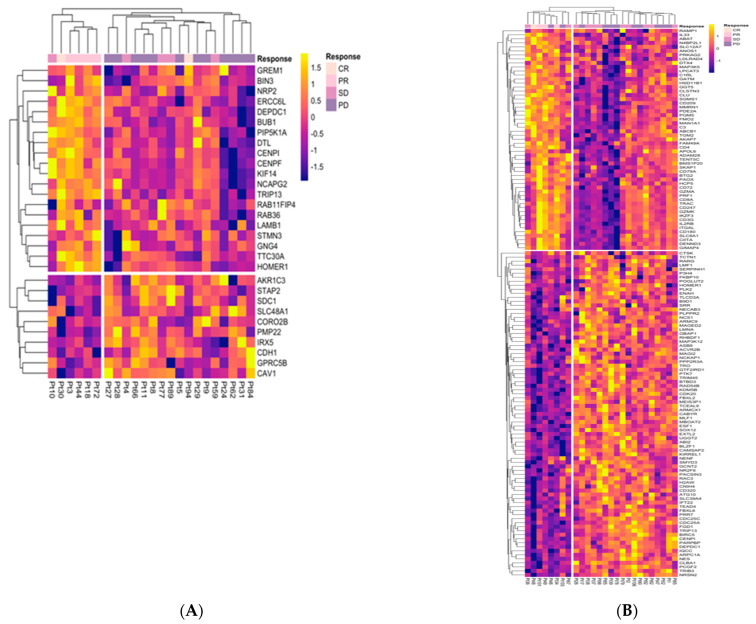
IMMUNETS genes stratify melanoma cohorts by response to nivolumab. Genes shown were found in at most two IMMUNETS networks and were differentially expressed between the response groups in (**A**) MEL_NAI (*n* = 23) and (**B**) MEL_PROG (*n* = 26). Clinical response is shown (top) and a cluster of responsive patients is found on the left of each heatmap. Blom-transformed gene expression is visualised on a yellow (highest) to blue (lowest) scale, thus lighter colours represent higher expression values.

**Figure 4 jpm-12-00958-f004:**
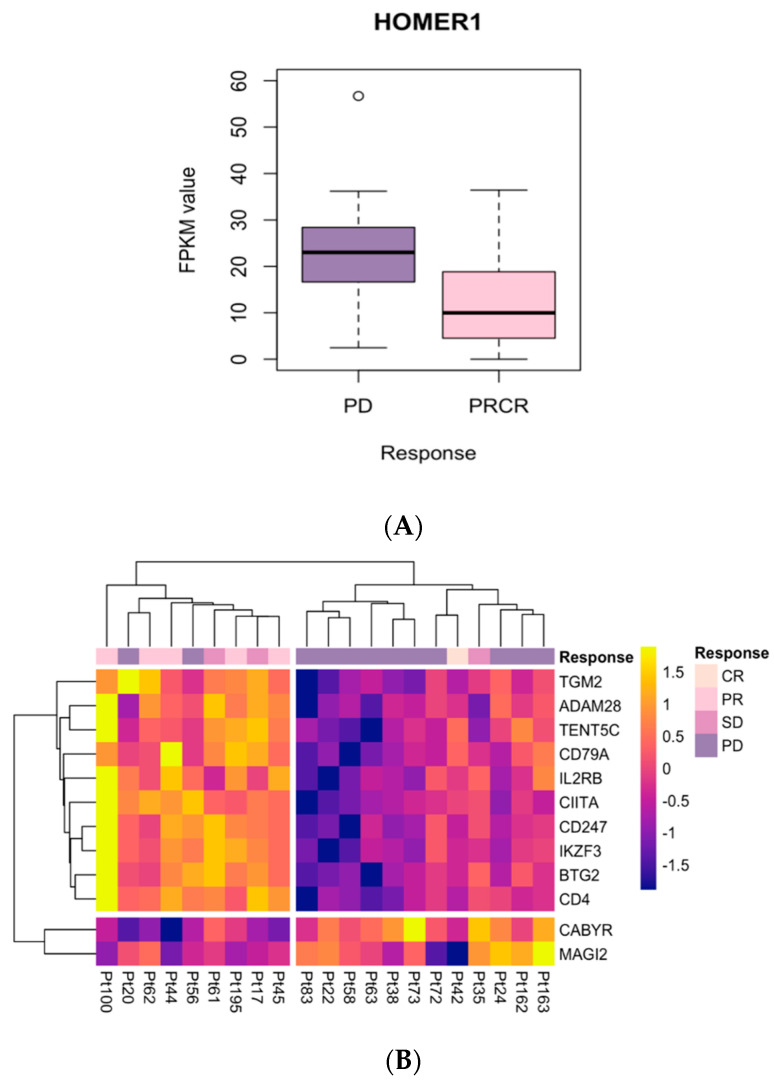
Candidate immune biomarkers from IMMUNETS are differentially expressed in independent datasets. (**A**): HOMER1 expression between responsive (PRCR, *n* = 15) and non-responding (PD, *n* = 14) groups in VALID_NAI. Significant expression differences were also observed in MEL_NAI, however the correlation with clinical response was not conserved, possibly arising from splice variation. (**B**) Twelve genes differentially expressed in VALID_PROG that were significant in MEL_PROG. The colour-coding for clinical response values is shown on the right-hand side. Heatmap colours represent Blom-transformed gene expression from yellow (highest) blue (lowest), values in the key represent a log scale.

**Figure 5 jpm-12-00958-f005:**
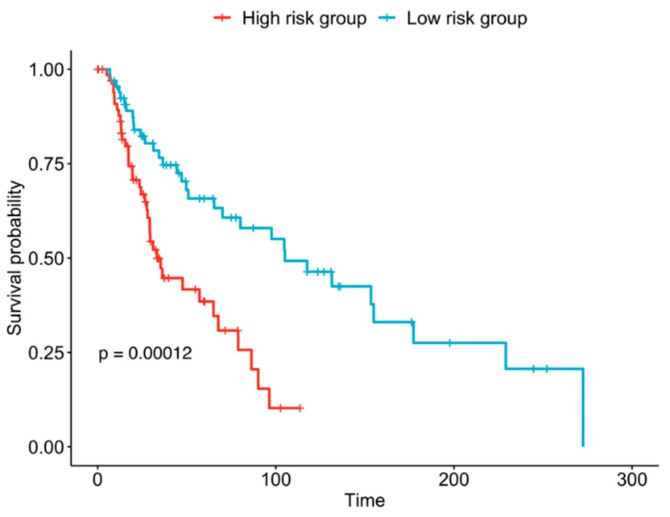
Performance of the four-factor prognostic model in blind test data (TCGA_VALID, *n* = 135). Melanoma patients were risk stratified by overall survival according to the variables Age, tumour stage, CIITA, and IKZF3. The *x*-axis shows time in months. The high-risk and low-risk groups have significantly different overall survival (*p* = 0.00012).

**Table 1 jpm-12-00958-t001:** Unique pairwise overlap between IMMUNETS cell types. Counts are shown for genes that are represented in only one (diagonal) or exactly two (off-diagonal) IMMUNETS networks. For example, the top left value (for T cell, T cell) identifies 233 genes present only in the T-net network, the value of 357, immediately below, corresponds to genes found in T-net and NK-net but not present in any other IMMUNETS network.

	T Cell	NK Cell	B Cell	Monocyte	Dendritic Cell
T cell	233	-	-	-	-
NK cell	357	935	-	-	-
B cell	131	535	1022	-	-
Monocyte	58	171	120	233	-
Dendritic cell	143	750	333	222	839

**Table 2 jpm-12-00958-t002:** Summary of DAVID clusters for immune-regulated, differentially expressed genes in MEL_NAI (*n* = 30) and MEL_PROG (*n* = 136). Clusters with significant enrichment score (≥1.3) are shown.

Dataset	Biological Descriptor (s)	Score	Genes
MEL_NAI	Cell cycle, Cell division, Mitosis	1.90	BUB1, ERCC6L, CENPF, CENPI, NCAPG2, KIF14, BIN3, DTL, DEPDC1, PIP5K1A, CDH1, AKR1C3, STAP2
MEL_PROG	Immunoreceptor signaling, ITAM	2.17	CD247, CD79A, CD3G, CD72, CD4
MEL_PROG	Immunity, Adaptive Immunity	1.81	CD180, CD79A, SKAP1, CD4, CD209, C1RL, MAP3K5, CD8A, C3, CLU
MEL_PROG	Regulation of immune response, including T cell receptor signaling	1.79	CD247, SKAP1, CD4, CD8A, CD3G, PRF1, ITGAL, TRAC, C3, CD72, CD79A, KIRREL1, PTK7
MEL_PROG	Immunity, Innate Immunity	1.44	CD180, CD79A, SKAP1, CD4, CD209, C1RL, MAP3K5, CD8A, C3, CLU
MEL_PROG	Complement pathway	1.38	C1RL, C3, CLU, CD180, CD209, MAP3K5
MEL_PROG	Antigen processing and presentation	1.33	CIITA, CD79A, CD4, CD8A, GZMA, C3

**Table 3 jpm-12-00958-t003:** Univariate risk stratification of MEL_TCGA (*n* = 390) and MIXED_ICI (*n* = 174) by overall survival with BIO_13. Log-rank test q-values are shown for risk stratification with groups defined by regularised Gaussian mixture modelling (please see methods). Asterisks (*) indicates *q*-value < 0.05.

Gene	MEL_TCGA	MI × ED_ICI
ADAM28	1.161 × 10^−4^ *	1.230 × 10^−2^ *
TGM2	2.368 × 10^−3^ *	4.607 × 10^−1^
CD247	8.566 × 10^−5^*	3.450 × 10^−2^ *
CD4	1.161 × 10^−4^ *	3.757 × 10^−1^
IKZF3	4.098 × 10^−6^ *	9.800 × 10^−4^ *
TENT5C	2.593 × 10^−4^ *	2.015 × 10^−1^
BTG2	4.684 × 10^−2^ *	4.607 × 10^−1^
HOMER1	9.643 × 10^−2^	1.055 × 10^−1^
CIITA	2.150 × 10^−4^ *	6.800 × 10^−4^ *
CABYR	5.233 × 10^−1^	2.015 × 10^−1^
CD79A	1.617 × 10^−3^ *	6.800 × 10^−4^ *
IL2RB	1.161 × 10^−4^ *	2.670 × 10^−3^ *
MAGI2	1.018 × 10^−1^	5.875 × 10^−1^

**Table 4 jpm-12-00958-t004:** Cox proportional hazards model for overall survival in the training data (TCGA_TRAIN, *n* = 255). Asterisks (*) indicates *p*-value < 0.05. This model was taken forwards for validation in TCGA_VALID (Figure 5).

Prognostic Factor	*p*-Value	Hazard Ratio	95% Confidence Interval
Age	0.00492 *	1.0189	1.00–1.03
Tumour stage	0.017 *	1.3205	1.05–1.66
CIITA	0.012 *	0.8602	0.76–0.97
IKZF3	0.976	1.0015	0.91–1.11

## Data Availability

The IMMUNETS networks, focus networks and BiNGO results are available as Appendix A. Immune Response In Silico (IRIS) data are available on GEO database (GSE22886). RNA-seq data from Riaz et al. study can be downloaded from GEO database (GSE91061). RNA-seq data from Liu et al. study is provided as Appendix A to the published study or can be downloaded from database of Genotypes and Phenotypes (dbGaP) with accession number phs000452.v3.p1. RNA-seq data of TCGA skin cutaneous melanoma patients are available on GDC data portal (https://portal.gdc.cancer.gov/). Data of MIXED_ICI cohort can be downloaded from dbGaP with accession number phs002683.v1.p1 for Freeman S.S. et al. study, phs000452.v3.p1 for van Allen et al. study; from GEO with accession number GSE78220 for Hugo et al. study; from the European Nucleotide Archive (ENA) with accession number PRJEB23709 for Gide et al. study. The processed data of all four studies in MIXED_ICI cohort are provided in Freeman S.S. et al. study at https://zenodo.org/record/5528497.

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
