# Peer review of "Immune Cell Networks Uncover Candidate Biomarkers of Melanoma Immunotherapy Response"

_jpm, 2022, doi:10.3390/jpm12060958_

Round 1

Reviewer 1 Report

Duong et al, here proposed an immune cell network system for cancer immunotherapy response projection, which is of high interest, both scientifically and clinically. However, several questions need to be addressed before consideration of potential publication.

  1. The authors claimed their IMMUNETS model provide candidate biomarkers with prognostic value for ICI treatment, more specifically, nivolumab in melanoma treatment. It appears to the reviewer that the presented work arrived at a conclusion that Age, Tumor stage, CIITA and IKZF3 are the 4 most prominent biomarkers. Where is the validation of this 4-factor model? To what extend does the IMMUNETS’s prediction align with real word data in terms of prognostic outcome.

  1. The authors excluded the expression of PD-L1 in their model (page 9). However, PD-L1 expression is repeatedly affirmed as a biomarker for predicting the efficacy of anti-PD1 antibodies, especially in melanoma, which happens to be the training dataset of this work. Please explain further of the rationale of the PD-L1 exclusion. Also, please comment on the superiority of the IMMUNETS over the PDL1 expression as prognostic biomarker.

  1. Again, PD-L1 as a prognostic biomarker can be applied to multiple cancer indications. To what extend can the IMMUNETS be applied to a broader cohort of melanoma patients as well as other cancer indications?

Reviewer 2 Report

In the manuscript entitled “Immune cell networks uncover candidate biomarkers of cancer immunotherapy response” the authors investigate the utility of novel biomarkers that determine the prescription of the immune checkpoint inhibitors to needy patients in the context of advanced melanoma cohorts and precision medicine. Below are the comments to improve the manuscript.

  1. Did the authors further corroborate the prognostic value of their 11candidate immune genes in the actual clinical settings? This is required as a further validation of their in silico
  2. The authors should also try to further validate these candidate genes in different contexts such as single ICI treatment, combination ICI treatments, ICI resistant patients etc.
  3. Is it possible that these genes be monitored in a large cohort of melanoma patients with diverse genetic backgrounds? The authors should discuss these situations.

Round 2

Reviewer 1 Report

The authors have elegantly addressed my comments